# Participatory mental health interventions in low-income and middle-income countries: a realist review protocol

Cheyann J Heap,[1] Hannah Maria Jennings ![ORCID],[2,3,4] Kaaren Mathias,[5,6] Himal Gaire,[7] Farirai Gumbonzvanda,[8] Nyaradzayi Gumbonzvanda,[9] Garima Gupta,[6] Sumeet Jain,[10] Bidya Maharjan,[11] Rakchhya Maharjan,[11] Sujen Man Maharjan,[11] Pashupati Mahat,[12] Pooja Pillai,[6] Martin Webber ![ORCID],[13] Jerome Wright,[2] Rochelle Burgess[3]

For numbered affiliations see end of article.

**Correspondence to**
Dr Hannah Maria Jennings;
hannah.jennings@york.ac.uk

## ABSTRACT

**Introduction** The launch of the Movement for Global Mental Health brought long-standing calls for improved mental health interventions in low-and middle-income countries (LMICs) to centre stage. Within the movement, the participation of communities and people with lived experience of mental health problems is argued as essential to successful interventions. However, there remains a lack of conceptual clarity around 'participation' in mental health interventions with the specific elements of participation rarely articulated. Our review responds to this gap by exploring how 'participation' is applied, what it means and what key mechanisms contribute to change in participatory interventions for mental health in LMICs.

**Methods and analysis** A realist review methodology will be used to identify the different contexts that trigger mechanisms of change, and the resulting outcomes related to the development and implementation of participatory mental health interventions, that is: what makes participation *work* in mental health interventions in LMICs and *why*? We augment our search with primary data collection in communities who are the targets of global mental health initiatives to inform the production of a programme theory on participation for mental health in LMICs.

**Ethics and dissemination** Ethical approval for focus group discussions (FGDs) was obtained in each country involved. FGDs will be conducted in line with WHO safety guidance during the COVID-19 crisis. The full review will be published in an academic journal, with further papers providing an in-depth analysis on community perspectives on participation in mental health. The project findings will also be shared on a website, in webinars and an online workshop.

## Strengths and limitations of this study

► The review is strengthened by stakeholder involvement—focus group discussions with people with mental health problems and carers in low- and middle-income countries.

► The review is theory driven, using concepts of participation from academic literature and applied practice as a framework.

► The review is limited to published academic papers in English; there may be further insights available in grey literature and in other languages.

## INTRODUCTION

Globally, mental ill health is a leading cause of disability.[1] Sociostructural factors are increasingly recognised as causes and consequences of poor mental health.[2][3] In 2007, *The Lancet's* pivotal 'Global Mental Health' series highlighted global inequalities in mental health provision, particularly in low-income and middle-income countries (LMICs),[4] marking the beginning of what is now commonly referred to as the *Movement for Global Mental Health.*[5] The movement advocates evidence-based interventions, informed by a human rights approach, with the aim to scale up mental health services. However, these efforts operate primarily within an essentially biomedical framework that eschews attention to structural drivers of distress in favour of pragmatics in clinical care. The movement has faced extensive critique because of this approach, with a special section of *Transcultural Psychiatry* in 2012 arguing for a greater emphasis of engagement with local communities within mental health to overcome these limitations.[6] For example, when communities actively participate in the governance, design or delivery of mental health interventions, they can be more acceptable and relevant to local needs as well as more cost-effect and higher quality.[7] Furthermore, depending on the social processes involved, participation can be empowering and even transformative



to communities.[8 9] However, an inattention to power dynamics when working on 'participation' run the risk of exacerbating existing patterns of exclusion and reinforcing inequities.[8]

As a group of researchers and practitioners, we are interested in the theory and application of participation in global mental health interventions in LMICs for two key reasons: first, we argue that how people in LMICs define participation in mental health interventions or more broadly matters to programmes. Second, we argue that participation in mental health interventions creates platforms for transformation in the lives of people it seeks to benefits *if* attention is paid to key processes.

## Theories of participation

Debates around the active involvement of communities in localities in LMICs—referred to as 'participation', 'community engagement' and 'mobilisation'—are important and related principles can be found within the broader fields of global health and international development. The diversity of approaches relating to the term also lead to its evolution into what some argue as a benign concept, a term with wide reaching boundaries that enable its wide-ranging applications.[10] There are different perspectives on how 'participation' is achieved and for what purposes participation is implemented.[9] For example, Campbell and Jovchelovitch[11] suggest that participation is a mechanism for negotiating social identity, and participatory power thus emerges as 'a space of possible action'. Others suggest that mental health 'consumer participation' must include the power to take meaningful action through orchestrating changes in service planning, delivery and intervention.[12 13] What action looks like is dependant on the specific theoretical approach applied in a certain context.

Participatory theories can be broadly separated into *typologies* (categories, often set as 'stages') or *continuums* (spectrums). 'Ladder' or stage typologies are common in civic, political understandings of power. Arnstein's[14] famous (1969) ladder of citizen participation has eight rungs from *manipulation* to *citizen control*. Similarly, Goetz and Gaventa[15] range from *consultation* (information sharing) to *influence* (tangible impact on policy making and service delivery), and Wilcox[16] considers collaboration between citizen and state from *information* to *initiatives*. Pretty frames 'participation' as opportunities for the poor to gain benefits, from *manipulative* to *self-mobilisation*,[17] whereas Eyben's[18] rights-based, six-rung ladder (*instrumental participation* to *participatory rights*) suggests that the right to participation facilitates access to other human rights. These ladders/stages have several commonalities: on lower rungs people have little power, and any inclusion of their voices is tokenistic and does not lead to tangible change. The top rungs are interventions designed and led by the community. However, ladders do not effectively accommodate the dynamic, contextual and relational aspects of power in participation. Additionally,

a fully citizen-led approach may not always be possible or even desirable.

Continuums can accommodate shifts in participation and power status. In community development, Draper *et al*[19] created a process-based continuum of participation across five categories including external support, leadership and women's empowerment. Bebbington and Farrington's[20] continuum focuses on participation breadth (representativeness of the full community) and depth (extent of meaningful involvement). Similar to participation ladders, Hickey and Kipping's[21] mental health participation continuum ranges from *information* to *user control*, with a shift from passive 'consumerism' to 'democratisation'. Likewise, Rifkin and Kangere[22] argue participation ranges from people as passive receivers to active decision makers, acknowledging change over time.

Some approaches emphasise the dialogical nature of participation. Campbell and Jovchelovitch[11] take a Freirean approach, noting how transformative dialogue can lead to reciprocal growth and learning between 'the powerful' and 'the community'. Wallerstein also highlights the capacity for mutual learning and reciprocal relationships, with partnerships between resource holders and beneficiaries leading to transformational change for both.[23] The South Asia Perspective Network Association goes further to emphasise deliberately 'pro-poor' strategies, where through a dialogical approach, sustained participatory action and new social movements lead to wider social transformation.[24] Nastasi *et al*'s[25] Participatory Intervention Model applies this idea to mental health interventions, noting the importance of relationships, changing interpersonal and power dynamics, and matching knowledge and support to the needs of the community. Indeed, communities are constantly shifting and have their own internal power dynamics.[26] White takes account of this in a relationally based typological approach, which accounts for shifting priorities and dynamics between the community and those with power.[27] This typology ranges from *nominal* (tokenistic involvement) to *transformative* (empowerment).

Online supplemental file 1 shows our mind map bringing together key debates in participation based around who it is for, how it is used and the impact (if any) on participatory approaches. The mind map is intended to be comprehensive but not exhaustive. It was supplemented by discussion on the lived experiences of practitioners in the team and the participatory work of their organisations.[28 29] We conceptualise 'participation' in mental health as *the active involvement of people affected by interventions or targeted action (including wider research projects)*. This can include a range of actors: service users, carers, providers and wider community members. Ultimately, they all contribute to the intervention in some capacity, and no group are merely passive recipients of an outcome. This includes involvement and ownership over various phases of concept, design, implementation and evaluation stages of interventions. Notably, our approach recognises a range of dynamics and factors that shape

participation including: interests and motivations, socio-economic context, influence, and structural and interpersonal power.

## Realist review on participation on mental health in LMIC

To date, there have been reviews of community engagement and participatory health approaches in LMICs[30 31] including an ongoing realist review on community engagement in non-communicable disease interventions and research[32]; however, to the authors' knowledge, there has been no specific review that interrogates participatory approaches to mental health across LMICs. The current review seeks to address this gap exploring the following questions: In LMIC settings, what is the nature of participatory approaches for mental health improvement? Who are the targets of participation in mental health research and practice? What factors contribute to their success or failure?

Our investigation applies a realist methodology, which has a similar interest in 'what works for these particular people in particular contexts'.[30 33] Realists reviews combine theory and empirical observation and seek to understand how and why change is facilitated.[30 32–34] This review will be used to identify mechanisms of change and their related contexts, which will support the development of theory to further our understandings of how participation is used in the design and implementation of mental health interventions.

## Aim and research questions

The aim of the review is to interrogate existing literature relating to participatory mental health interventions in LMICs, to consider how, for whom and under what conditions participatory approaches work. Across two stages of data extraction, we will seek to answer the following questions:

1. Why and for whom has participation been used in mental health interventions in LMIC?
2. How and to what extent has participation been operationalised in research versus implementation?
3. What are the mechanisms of action of participation and how are they linked to local contexts?
4. Why, how and under what circumstances does community participation in mental health interventions lead to improved mental health?

## METHOD

This review was designed adapting Saul *et al*'s suggested 10 steps for a realist review.[35] Realist reviews often involve developing an initial programme theory (IPT), which is later refined through the development of middle-range theories of what works.[35–37] Middle-range theories are general enough to be transferred to other projects but concrete and specific enough to apply to practice.[33] In generating or exploring theory, context, mechanism and outcomes (CMO) are crucial (figure 1: CMO configurations). When relations between these factors are explored,

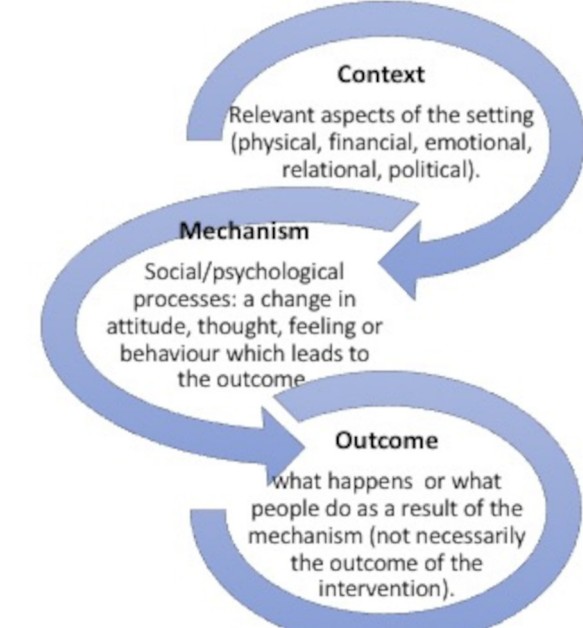

**Figure 1** CMO configurations.

they create 'CMO configurations' and are developed in response to particular research questions.[37]

As not all papers will have enough detail to infer theories and extract CMO configurations, data extraction will be in two stages. Stage 1 will extract surface-level study characteristics and quality. Stage 2 will complete a deeper analysis of the highest quality papers, from which CMO configurations will be extracted to build our middle-range theories. To counter the western hegemonic knowledge structures that shape much of the peer-reviewed literature,[38] we have added specific stakeholder consultations in our review, to combine and include embodied knowledges held by people with mental health difficulties, carers and community members. Consulting stakeholders as part of the review is consistent with the realist methodology.[30 32] We assess their views of participation and involvement as it relates to mental health, at two stages in the review . The 10 steps as we have applied them in our study are described in more detail further (figure 2: 10-step review process).

We will report the results of the review according to the Realist and Meta-narrative Evidence Syntheses: Evolving Standards quality and publication standards.[33]

## Concepts of 'participation' and developing research questions (steps 1 and 2)

A mind map of different concepts of 'participation' (online supplemental file 1) was created between January and March 2021. This involved reading research literature on participation, particularly on LMICs and from the academic fields of development, health and human rights. This was supplemented by monthly meetings between the entire research group to discuss emerging themes and link the research to the applied practice of



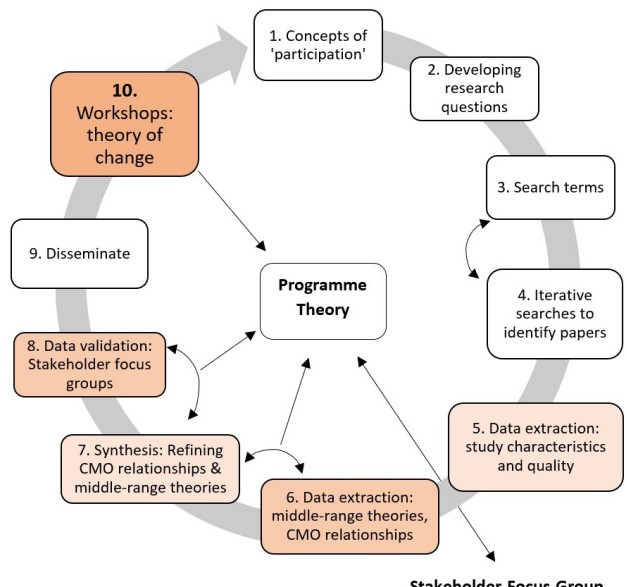

**Figure 2** Ten-step review process. CMO, context, mechanism and outcome.

team members. Team members, across LMICs and high-income countries, represent varied ethnic backgrounds and disciplinary backgrounds in mental health, medicine, social work, mental health nursing, psychology, academia and non-governmental organisations. Key debates from the mindmap and group discussion were used to develop the four research questions.

Based on this initial conceptual review and discussions, project principle investigator (RB) drafted an IPT and model that was discussed and refined within the research group (figure 3: programme theory). The IPT guided the remainder of the review process and will be refined in light of the findings of this review (see step 9).

Our IPT considers contexts at multiple levels and is purposefully broad. The IPT suggests, that overall, individuals who live with or are at risk of developing mental health conditions in the global south (LMIC) (who) do so because of a number of daily realities across various settings with relations to mental health needs (contexts barriers). We postulate that these wider limiting contexts will include: poor mental health service infrastructure; low uptake of services due to low awareness of services and due to a poor 'fit' of services to the needs of communities; stigma and exclusion faced by people living with mental health issues; histories of silencing of lived experience within the mental health space; and intersecting social, political and environmental challenges that link to mental health conditions. The second level of contexts we are interested in are those that may contribute positively to the impact of interventions and mental health (enabling contexts). These include various forms of capital, agency, collective mobilisation and supportive community partnerships.

We believe that if participatory approaches to mental health meet the parameters of *meaningful participation* (which we defined as transformative participation outlined by White's[27] framework) then this would activate a series of mechanisms that would lead to positive outcomes on mental health conditions, as well as a wider impacts on societal factors linked to poor mental health (addressing hindering contexts). We believe this would be the case across various population groups at risk for developing mental health conditions (who). This is because transformative participation creates opportunities for participants to change the environments that place good mental health at risk, as well as increase access to better services through various pathways such as the production of new communities of practice and support networks.

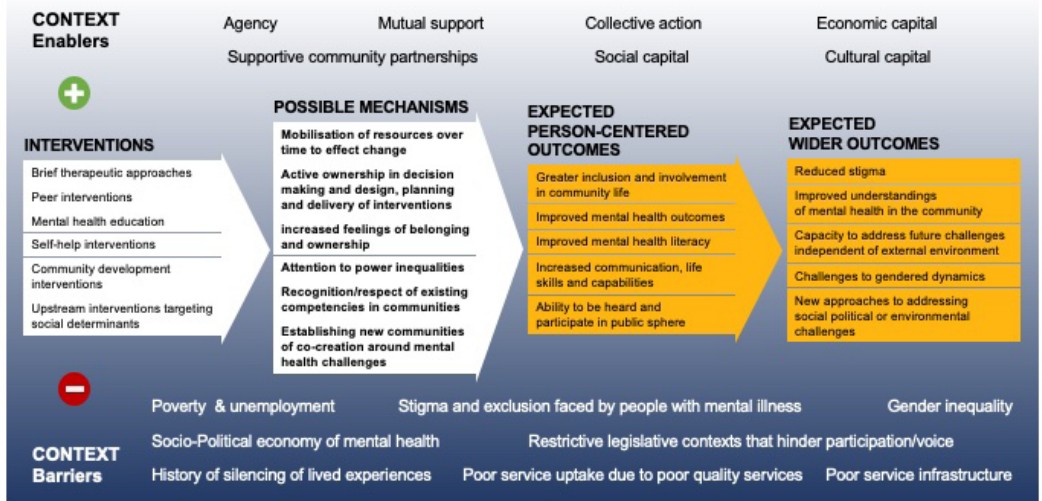

**Figure 3** Programme theory.

We imagine the potential outcomes could occur at two levels. At the individual level, this may include: improved mental health outcomes; improved access to services, improved mental health literacy and a reduction in experiences of exclusion. At wider societal levels, it could include: a reduction in harmful social, political and environmental drivers of distress (ie, reduced poverty; reduced gender discrimination; improved living conditions) and increased voice and ownership for people with lived experience in wider society. We believe that when meaningful participation (transformative) is not present, positive impacts may occur but only for individual outcomes. Ultimately, this theory helps us to determine not only the mechanisms through which participatory interventions may work, but the ability for participatory interventions to be activated in the presence of enabling contexts. Also, our IPT allows us to account for the fact that transformative participatory interventions themselves may work to change the contexts that trigger or block the mechanisms that drive better mental health. As such, we do not imagine a hard line between interventions and mechanisms but will allow our literature review and stakeholder engagement to help illuminate these dynamics in the contexts of interventions that have transformative interests at their core.

### Search terms and identifying papers (steps 3 and 4)
Search terms and databases searched were discussed in team meetings in early 2021. Searches are being carried out in the databases: Scopus, Medline/PsychINFO, ASSIA, CINAHL, Embase and JSTOR. If additional studies are identified in the references or through the team, they will be considered for screening. Table 1 shows our search terms. Online supplemental file 2 gives details for the full search strategy.

### Screening and study selection
Instead of limiting to a particular definition of 'participation', this review will take a pragmatic approach. It will include all papers which either: (A) claim to be participatory or (B) clearly include participation (the active involvement of people with mental health problems and their supporters). We will include studies located in LMICs (as according the World Bank definition) that are related to a mental health intervention. We use a broad definition of mental health as defined by the WHO and draw on their list of mental and neurological disorders.[39] We define interventions as programmes or policies that are implemented with the aim to change outcome/s. Table 2 shows our inclusion and exclusion criteria.

Possible relevant papers identified during the searches will be uploaded onto the software Ryaan for title and abstract screening. Title and abstract screening will be conducted blinded by members of the research team to ensure consistency 10% of studies screened by two members of the research team. Full-text screening will be completed by the whole research team with members working in pairs and the papers divided among the team. Any uncertainties throughout the process will be discussed within the team at regular team meetings.

### Data extraction and quality appraisal (steps 5 and 6)
There will be two stages of data extraction. The first stage will answer the first two research questions of the review.:
1. Why and for whom has participation been used in mental health interventions in LMIC?

| Table 1 | Literature search terms | | |
|---|---|---|---|
| The overall search formula included three umbrella concepts: (('*Participation*') AND ('*Mental health*') AND ('*LMICs*')). The terms under each umbrella (eg, 'Participation') were separated by the Boolean operator *OR* | | | |
| | 'Participation' | 'Mental Health'* | 'LMICs' (low-income to middle-income countries) |
| | ▶ participant<br>▶ empower<br>▶ community<br>▶ community?led<br>▶ co-designed<br>▶ inclusion<br>▶ inclusive<br>▶ capacity building<br>▶ capabilities<br>▶ engagement<br>▶ consultation<br>▶ co-produc<br>▶ peer-led<br>▶ peer?to?peer<br>▶ task?shifting<br>▶ task?switching | ▶ psychological disorder<br>▶ psychological problem<br>▶ psychological illness<br>▶ psychological distress<br>▶ psychiatric disorder<br>▶ psychiatric problem<br>▶ psychiatric illness<br>▶ psychiatric distress<br>▶ pychosocial disability<br>▶ mental illness<br>▶ mental health<br>▶ serious mental illness | ▶ Developing countr<br>▶ low?income countr<br>▶ middle?income countr |

Phrases of more than one word were put into 'double quotes' for the search.
*There are a wide range of conceptions of mental health, including medical ('mental illness') and psychosocial.[39] We tried to accommodate this variety in our search terms.

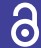

**Table 2** Inclusion and exclusion criteria

| Inclusion criteria | Exclusion criteria |
|---|---|
| Located in low-income and middle-income countries; health and development projects/ interventions addressing mental health; intervention, or research study about an intervention, which claims to be participatory OR clearly demonstrates the active involvement of the target population in concept, design, implementation or evaluation; any 'level' of intervention including individual, group and systemic interventions such as national and international projects; any method – qualitative, quantitative, case study; participants are people with mental health problems, their unpaid carers or other people from the community such as local health workers or laypeople; on or after 2001, based on the World Health Report that presented community-based mental health services as critical.[42] | 'Participation' does not include the active involvement of people from the target community (eg, laypeople, local professionals, people with mental health problems and carers); not about an intervention ('intervention'=support group, a specific mental health policy, training) |

2. How and to what extent has participation been operationalised in research versus implementation?

At the end of the first stage, the highest quality papers with the most detail about theories of participation and CMOs will be identified and taken forward to the second stage. This more in-depth analysis and CMO configuration extraction will answer the final two questions:

3. What are the mechanisms of action of participation and how are they linked to local contexts?
4. Why, how and under what circumstances does community participation in mental health interventions lead to improved mental health?

### Stage 1

As with the full-text screening, data extraction will be completed in pairs by the research team. A data extraction tool will be developed and refined as relevant data are extracted. Extracted data will likely include: study aims, methods, why participation is used; concept and theory of 'participation'; target population and stakeholders; how participation is 'done'; outcomes of intervention; level of participation; and analysis of power. For quality, a realist review asks whether papers are *relevant* (to the research question) and *rigorous* (of good-enough quality to make a meaningful contribution).[33] The data extraction tool will include whether papers are low, medium or high relevance. For 'rigour', as we are including all methods (case studies, qualitative and quantitative), specific quality checklists will be selected based on the nature of the papers, primarily drawing from the Joanna Briggs Institute Critical Appraisal tools.[40] Uncertainties as to the data extraction, relevance and rigour will be discussed in the team meetings and research associate (CJH) and principle investigators (RB and HMJ) will do final quality checks.

### Stage 2

Papers of 'high quality'—rigorous, relevant and with sufficient information to extract CMO configurations—will be taken forward for the second stage of review. Potential studies for this stage of data extraction will be identified by the research team during stage 1. All studies identified and those marked as 'high' for relevance will be considered. Three members of the research team (CJH, RB and HMJ) will read the shortlisted papers and agree through consensus whether they should go through to the next round of the review. They will record and report back to the wider team the reasoning for the decisions. The criteria for studies to go through to the next round of review include: information and discussion about participation, enough detail to extract information on CMO configurations.

### Data synthesis and CMO configurations (step 7)
#### Stage 1
The first stage of data synthesis based on the first round of data collection will be narrative, based around the first two research questions. It will report on the breadth and scope of participation as used in mental health settings in LMICs. Comparisons will be made with the FGDs and their understanding of participation.

#### Stage 2
The second round of data synthesis will be a theory focused iterative process based on the 'high quality' papers'. Data synthesis will take the following process:

► Taking our 'high quality' papers, we will extract CMO configurations related to participation from each paper. This will be an iterative process where researchers will work in groups to discuss and extract the configurations. They will consider the evidence for the relationships between CMO relating to participation before finalising the CMO configurations.

► CMO configurations will be combined and refined to identify 'enabling contexts', the types of 'enabling mechanisms' they trigger and their possible outcomes (midrange theories). Again, this will be an iterative process involving the whole network.

► Based on these findings and the FGD analysis, we will revise our IPT.

We plan to undertake this process as a team, involving workshops led by KM (who has experience in realist methodology).[41] One workshop will develop our initial CMO configurations based on the 'high quality papers'. Subsequent team meetings will allow refining of CMO configurations, comparing them against a thematic analysis of FGDs. As theories and CMO configurations are gradually refined, the process will be carefully documented to justify changes and ensure a rigorous and transparent process. Overall, the synthesis should be such that the weaknesses (or omissions) in one particular paper from the review are accommodated for by the strengths of other papers. Once the review has been completed, we will have a set of middle-range theories, informed by the embodied knowledge of everyday citizens who are often the targets of mental health interventions in LMICs. The theories explain how participatory mental health interventions create changes in local context (C), to trigger sociopsychological mechanisms (M) that lead to new behaviour outcomes (O) in communities of interest. Final CMO configurations will be used to refine the IPT by RB and HMJ, which will be feedback and refined by the wider group.

### Focus group discussions (step 8)

There will be two rounds of stakeholder consultations through focus group discussions, across four sites in three countries (Nepal, India and Zimbabwe), organised by members of the network who work in participation and mental health (BM, RM, SMM, PM, PP, NG, FG and GG). The groups will include people affected by mental health difficulties, carers and community members.

The first round of FGDs will explore understandings of participation and contribute to the IPT. In the second round of FGDs, a summary of data so far from the literature review will be presented in order to get feedback and reflections from the participants and will further contribute to the refinement of the programme theory. Findings of the FGDs will be compared and integrated with findings from the data extraction to help theorise 'what works and how' in the way that is considered most participatory in the context. Findings from the FGDs will be reported in our final write-up, dissemination and revision of our IPT (steps 9 and 10).

### Dissemination and workshop (steps 9 and 10)

It is anticipated that the current realist review will create a strong theoretical and practice-based foundation for future work in participatory research and practice for mental health in LMIC settings and other environments of adversity. It will also directly shape the efforts of our network for participatory mental health in LMICs. We will disseminate our findings within a workshop series; this will allow us to expand our network by welcoming a range of interested stakeholders including people with mental health problems, paid and unpaid carers, practitioners, academics and policymakers. We also plan to disseminate our findings though peer-review papers and blogs written in accessible and local languages in each of our country settings.

### Patient and public involvement

As part of the review, we will have planned stakeholder consultations with people with mental health difficulties, carers and community members in three LMICs (Nepal, India and Zimbabwe). The consultations are an important part of the review and will contribute to our refined programme theory. As part of our dissemination, we will organise workshops with people with mental health difficulties, carers, practitioners and community members in LMICs.

### ETHICS AND DISSEMINATION

Ethical approval for the study received from the University of York Health Science and Research Governance committee (HSRGC/2021/438) and we have received ethical approval from the relevant bodies of the countries involved (Nepal Health Research Council, 3026; Emmanuel Hospital Association Institutional Ethics Committee, 254; UCL Ethics Board 127/002 and The Women's University in Africa 02/2020). The review has been registered with PROSPERO (ID number CRD42021241787). The full review is intended to be published in an academic journal, with further papers providing an in-depth analysis on community perspectives on participation. The project findings will also be shared on a website, in webinars and an online workshop.

#### Author affiliations
[1] Department of Social Work and Social Policy, University of York, York, UK
[2] University of York, York, UK
[3] UCL Institute for Global Health, London, UK
[4] Hull York Medical School, Hull and York, UK
[5] School of Health Sciences, University of Canterbury, Christchurch, New Zealand
[6] Burans, Herbertpur Christian Hospital, Uttarakhand, India
[7] Centre for Mental Health Counselling (CMC), Katmandu, Nepal
[8] Rozaria Memorial Trust, Harare, Zimbabwe
[9] Rozaria Memorial Trust, Harare, UK
[10] School of Social and Political Science, The University of Edinburgh, Edinburgh, UK
[11] Chhahari Nepal for Mental Health (CNMH), Katmandu, Nepal
[12] Centre for Mental Health and Counselling, Katmandu, Nepal
[13] Department of Social Policy and Social Work, University of York, York, UK

**Contributors** RB and HMJ initiated the review. All members of the network contributed toward the mind-map and conception of the protocol. Following discussions with the whole team CH designed and drafted the initial protocol, which was subsequently reviewed by BM, KM, PM, MW, RM, SM, JW, HMJ and RB. BM, PM, KM, GG, PP, FG, NG and CH have taken a lead in planning the FGDs. HMJ, RB, BM, PM, KM and PP have organised ethical approval in respective sites. RB and HJ finalised the protocol. All authors approved the final manuscript (CH, HMJ, HG, FG, NG, GG, SJ, BM, RM, SM, PM, KM, PP, WM, JW, RB).

**Funding** This work was supported by Global Challenges Research Fund Pump Priming Award funded through the UKRI-MRC (2019-2020/2020-21 H0027804).

**Competing interests** None declared.

**Patient and public involvement** Patients and/or the public were involved in the design, or conduct, or reporting, or dissemination plans of this research. Refer to the Methods section for further details.

**Patient consent for publication** Not applicable.

**Provenance and peer review** Not commissioned; externally peer reviewed.

**ORCID iDs**
Hannah Maria Jennings http://orcid.org/0000-0002-8580-0327
Martin Webber http://orcid.org/0000-0003-3604-1376

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
