## [Reviewer comments · BMJ Open]

ARTICLE DETAILS

TITLE (PROVISIONAL)	Participatory mental health interventions in low and middle income countries: a realist review protocol
AUTHORS	Heap, Cheyann; Jennings, Hannah; Mathias, Kaaren; Gaire, Himal; Gumbonzvanda, Farirai; Gumbonzvanda, Nyaradzayi; Gupta, Garima; Jain, Sumeet; Maharjan, Bidya; Maharjan, Rakchhya; Maharjan, Sujen Man; Mahat, Pashupati; Pillai, Pooja; Webber, Martin; Wright, Jerome; Burgess, Rochelle

VERSION 1 – REVIEW

REVIEWER	Gilmore, Brynne University College Dublin, UCD Centre for Interdisciplinary Research, Education and Innovation in Health Systems
REVIEW RETURNED	11-Nov-2021

GENERAL COMMENTS	Dear authors, thank you for the opportunity to review your manuscript detailing a RRR protocol. I thought a lot of attention has gone into the design and reporting, and it is an important research area which will benefit from your study. While the overall design of the study is appropriate, I think the manuscript has a lack of understanding or clarity around realist review methodology and the reporting needs strong revision. Intro: • I thoroughly enjoyed reading your work on participation. This presents a very detailed description of different ‘typologies’, in a clear and succinct manner.• Consider moving paragraph from pg. 9 line 5-30 and incorporate within the ‘Concepts of ‘participation’’ section in methods, as this is your result of Step 1. RRR on Participation • Why is a Rapid Realist Review chosen over a traditional Realist Review? The outlined protocol could likely be a full RR itself. I would suggest either revising to be a RR, or the authors need to better explain why a ‘Rapid’ review was chosen. The ‘Rapid’ does not just apply to timeframe, but the methodological adjustments you make to support expediting the review. (i.e. what steps along the review were adjusted to support a more expedited review?: fewer databases, only one screener, etc).• Pg. 9, line 47: this questions is not well aligned with realist methodology. Consider revising to a more ‘what works, for whom, and why...’ style of question• Pg. 9, line 59: I do not think that RRs work to ‘compare’ information necessarily.• Pg 10. Line 3: For realist studies, while ‘mechanism of change’ are essential to elicit, more important is the relationship with
---

context. Is it this relationship (C+M) that drives outcomes/change, and is therefore a key component of realist studies.

- Specific research questions –I am not convinced that these are questions supported by a realist approach. For question 1, the ‘why and how’ aspect of a realist review is not about what has been done, but what has worked (or not worked). So, more like “What has worked for participation in MH interventions, why did this work and for whom did this work (or not work). Similar issue with Q3. I understand that these form part of Stage 1 so not necessarily needing to be ‘realist’ but then these components are missing from the ‘realist’ part (the how, why and for whom).
- o Question 4 – what particular context? There will be numerous contexts, and this is part of your data. A key aspect of a realist study is to identify/elicit ‘generative causation’. Generative causation is the relationship of Contexts to Mechanism to Outcomes. Essentially – what contexts trigger what mechanisms, and together what outcome(s) does this relationship produce (i.e. CMOc). So, to be a proper realist study identifying this generative causation is key.

Method

- Pg. 10, line 51 – not ‘often’ involve. They do involve developing and refining theories. For this line also, middle range theories are often the ‘end product’ or desired product. But we don’t always start with them. Initial theories, programme theories etc, but be better suited here.

- Pg 10. Line 56 –Context Mechanisms Outcomes in themselves are not really known as configurations - it’s their relationship to each other that make them a configuration (i.e. generative causation, when a C triggers an M, and this makes an O).

- Pg. 11, line 3 – can you clarify “how participation happens in research”. Is the review looking at participation in mental health research, or in mental health interventions?

- Pg 11. Line 22 – inclusion of stakeholders great and important addition. This is also consistent with realist review methodology, as an inclusion of an ‘advisory group’ or the like. Consider adding reference to this.

- I think your IPT is missing mechanism. Often using the ‘if, then, because’ statements to relay IPTs is helpful, so consider adding the ‘because...’ (which highlights mechanisms). For instance (a more specific one but just an example): if in contexts with poor mental health infrastructure, people who live with mental health conditions have meaningful participation in interventions (C) then there can be improved access to services (O) because new communities of co-creation around mental health challenges

- Figure 2: MRTs are usually the end product of the review. You extract CMOcs, and refine into IPTs/PTs. Consider adding in the IPT in Phase 2, and show where/how this is refined throughout the study.

- Data analysis – Please detail how will you extract and analyse CMOcs?

- I think the FGDs are a great contribution, but I am not sure they are being utilised in the best way for the review. What contribution does the first round of FGDs make to the review? I think using this to develop your IPT would better support its development and the scope of the review, and is also more consistent with realist methodology. The second round may be best placed after the extraction of CMOc, where you can present refined PTs to the stakeholders who can work to make findings more contextually specific.

	 • FGDs – how are you conducting them (as in, are you using theory driven interviewing?). Why thematic analysis as opposed to a more realist style of analysis? • Stage two: why did you propose to extract 1-4 CMOc/paper? This is unusual and typically any included paper you'd extract all CMOc, which will vary depending on the relevance/robustness of the paper to the question. • Stage two: a key aspect of a CMOc is that it occurs as a unit. Identifying an enabling mechanisms doesn't say much without the context that triggers it, because this can changes across context, and their relationship produces the outcome. • The line “how participatory mental health interventions create changes in local context (C), to trigger socio-psychological mechanisms (M) that lead to new behaviour outcomes (O) in communities of interest.” Other:  • Please be consistent across the terminology: CMOcs or CMO relationships, IPT, PT or MRT etc. • In abstract, Intro – ‘respond’ to ‘responds’ • In abstract, Methods ‘LIMCs’ to “LMICs” • In abstract, first instance of FGDs please spell out. • In abstract, title LMICs as ‘low-and middle-income’ in introduction, they are ‘low-to-middle income’. For continuity can one term be used throughout. • Intro, line 20 – should ‘movement’ be capitalised to be “Movement for GMH”? • Intro, line 33 – comma after “For example” • Intro, Line 37 – awkward ending “...more cost-effective more and ...” • Methods, pg. 9 line 56 “Realist review” to “Realist reviews...” • Please align reporting to, or reference RAMESES 1.
--	--

REVIEWER	Klingberg, Sonja University of the Witwatersrand, SAMRC/Wits Developmental Pathways for Health Research Unit (DPHRU)
REVIEW RETURNED	22-Nov-2021

GENERAL COMMENTS	Thank you for the opportunity to review this manuscript. I enjoyed reading this well-developed protocol, and I think both the protocol and, eventually, the review itself will make very valuable contributions to the literature (and hopefully practice!) around participatory approaches to mental health promotion. I found this protocol to be generally well written and very well contextualised, including an effective overview of theories of participation and a valuable mind map tracing the development of the authors’ thinking around the topic. Specific comments and questions:  1. Apart from the limited timeframe, it is not fully clear to me why this extensive process, including valuable stakeholder engagement, “only” amounts to a rapid realist review and not a full realist review but the process is otherwise clearly described. No action needs to be taken in response to this comment but it does seem to me like you are doing more or less the same amount of work as a full realist review would require. 2. My main feedback concerns the initial programme theory as I am finding the text version of it a little difficult to follow. It seems to suggest participatory approaches are a condition of effective
---

	interventions? Does this imply something specific about meaningful outcomes as a result of meaningful participation, i.e. would positive outcomes of 'not meaningfully' participatory interventions not be as meaningful (e.g. not transformative or empowering), or are you theorising that only meaningful participation will actually result in effective interventions? This is naturally something the review itself will clarify and refine but it is difficult to follow the argument here. It would be helpful to clarify the text description of the theory as Figure 3 does not express it in quite such specific terms. It may also be relevant to consider Abimbola's interesting paper "Beyond positive a priori bias: reframing community engagement in LMICs." in this context. However, to be clear, I am not saying that the initial programme theory itself needs to change – it may just need to be rephrased and expressed more explicitly. 3. The relationship between Context and the other constructs in Figure 3 is not fully clear – is it a cross-cutting dimension feeding into all the others equally? 4. It also seems to me based on Figure 3 that you are proposing that Interventions are included in the CMO configurations as a separate construct as (CMIOc) as opposed to being expressed through mechanisms. It may well be very relevant to do so, but this is not described in the steps of the review so this aspect could also be clarified or elaborated even if just to say that it is a possibility and the review itself will determine its relevance. 5. Very minor comment: "Qualitative" is listed twice under inclusion criteria. Thank you!
--	--

VERSION 1 – AUTHOR RESPONSE

Reviewer 1:
Dr. Brynne Gilmore, University College Dublin

Comments to the Author:

Dear authors, thank you for the opportunity to review your manuscript detailing a RRR protocol. I thought a lot of attention has gone into the design and reporting, and it is an important research area which will benefit from your study. While the overall design of the study is appropriate, I think the manuscript has a lack of understanding or clarity around realist review methodology and the reporting needs strong revision.

Response: We thank the reviewer for their kind comments and helpful suggestions. We have responded to the suggestions made below and amended the manuscript accordingly. We believe it is now a much stronger paper.

Intro:

- I thoroughly enjoyed reading your work on participation. This presents a very detailed description of different 'typologies', in a clear and succinct manner.

Response: Thank you, we enjoyed writing it!

- Consider moving paragraph from pg. 9 line 5-30 and incorporate within the ‘Concepts of ‘participation’” section in methods, as this is your result of Step 1.

Response: We considered this suggestion. However, we feel it is more appropriate that the section remains as part of the introduction as it provides important context to the review. Additionally, there is some overlap between the ‘steps’ described in the methods and the introduction/background – the research questions were also developed as part of step one and they too are in the introduction.

RRR on Participation

- Why is a Rapid Realist Review chosen over a traditional Realist Review? The outlined protocol could likely be a full RR itself. I would suggest either revising to be a RR, or the authors need to better explain why a ‘Rapid’ review was chosen. The ‘Rapid’ does not just apply to timeframe, but the methodological adjustments you make to support expediting the review. (i.e. what steps along the review were adjusted to support a more expedited review?: fewer databases, only one screener, etc).

Response: Thank you for this comment. We initially planned to do this as a rapid realist review, but over time, this expanded and we agree that this was not conducted in a rapid methodology. We had multiple data screeners, deployed a large team and explored a wide range of databases in our screening of literature, over a significant amount of time. As such, we have changed this to a realist review methodology.

- Pg. 9, line 47: this questions is not well aligned with realist methodology. Consider revising to a more ‘what works, for whom, and why...’ style of question

Response: We have now revised these initial questions as suggested. They read as “what is the nature of participatory approaches for mental health improvement? Who are the targets of participation in mental health research and practice? What factors contribute to their success or failure?”

- Pg. 9, line 59: I do not think that RRs work to ‘compare’ information necessarily.

Response: We have deleted this sentence, as we have opted not to describe the study as rapid. The previous sentence describing the purpose of realist reviews is more in line with our interests.

- Pg 10. Line 3: For realist studies, while ‘mechanism of change’ are essential to elicit, more important is the relationship with context. Is it this relationship (C+M) that drives outcomes/change, and is therefore a key component of realist studies.

Response: Many thanks, we have now clarified this sentence.

- Specific research questions –I am not convinced that these are questions supported by a realist approach. For question 1, the ‘why and how’ aspect of a realist review is not about what has been done, but what has worked (or not worked). So, more like “What has worked for participation in MH interventions, why did this work and for whom did this work (or not work). Similar issue with Q3. I understand that these form part of Stage 1 so not necessarily needing to be ‘realist’ but then these components are missing from the ‘realist’ part (the how, why and for whom).

Response: We have now edited the realist ‘end’ of the research questions to reflect your concerns – they are now fewer in number and follow a more recognisable format that has a focus on context and

mechanisms.

- Question 4 – what particular context? There will be numerous contexts, and this is part of your data. A key aspect of a realist study is to identify/ elicit 'generative causation'. Generative causation is the relationship of Contexts to Mechanism to Outcomes. Essentially – what contexts trigger what mechanisms, and together what outcome(s) does this relationship produce (i.e. CMOc). So, to be a proper realist study identifying this generative causation is key.

Response: Thank you for prompting us to more fully engage with realist methods – we have rephrased the research questions to clarify our focus is on the relationship between contexts and mechanisms and how they do or don't lead to improved mental health.

Method

- Pg. 10, line 51 – not 'often' involve. They do involve developing and refining theories. For this line also, middle range theories are often the 'end product' or desired product. But we don't always start with them. Initial theories, programme theories etc, but be better suited here.

Response: We have amended the text to say realist review often start with a programme theory which is refined through mid-range theories of what works and how.

- Pg 10. Line 56 –Context Mechanisms Outcomes in themselves are not really known as configurations - it's their relationship to each other that make them a configuration (i.e. generative causation, when a C triggers an M, and this makes an O).

Response: We have amended the text to clarify that is the relationships between context, mechanisms and outcomes that make them configurations.

- Pg. 11, line 3 – can you clarify "how participation happens in research". Is the review looking at participation in mental health research, or in mental health interventions?

Response: in the revised research questions we have clarified that the review is looking at how participation has been operationalised in research versus implementation of interventions. We have deleted this line in the text here as the revised text clarifies the overall methodology of the review.

- Pg 11. Line 22 – inclusion of stakeholders great and important addition. This is also consistent with realist review methodology, as an inclusion of an 'advisory group' or the like. Consider adding reference to this.

Response: We have added references.

- I think your IPT is missing mechanism. Often using the 'if, then, because' statements to relay IPTs is helpful, so consider adding the 'because...' (which highlights mechanisms). For instance (a more specific one but just an example): if in contexts with poor mental health infrastructure, people who live with mental health conditions have meaningful participation in interventions (C) then there can be improved access to services (O) because new communities of co-creation around mental health challenges.

Response: We have now added this. We have adjusted to manuscript to add a detailed description of the IPT, in the format suggested.

- Figure 2: MRTs are usually the end product of the review. You extract CMOcs, and refine into

IPTs/PTs. Consider adding in the IPT in Phase 2, and show where/how this is refined throughout the study.

Response: We have now adjusted figure 2 to include the programme theory and the different stages that will contribute to its development and refinement.

- Data analysis – Please detail how will you extract and analyse CMOcs?

Response: We are extracting and analysing CMOs during the second stage of data synthesis. Further details have been added under step 8 and 9 (data synthesis and CMO configurations) stage 2.

- I think the FGDs are a great contribution, but I am not sure they are being utilised in the best way for the review. What contribution does the first round of FGDs make to the review? I think using this to develop your IPT would better support its development and the scope of the review, and is also more consistent with realist methodology. The second round may be best placed after the extraction of CMOc, where you can present refined PTs to the stakeholders who can work to make findings more contextually specific.

Response: The FGDs (first round) have helped us to develop our IPT. The second round will reflect on some of the findings from the literature review and will contribute further to the refinement of the IPT. We have adjusted figure 2 to reflect this, and this has been explained in the manuscript.

- FGDs – how are you conducting them (as in, are you using theory driven interviewing?). Why thematic analysis as opposed to a more realist style of analysis?

Response: For the FGDs we were keen to follow an inductive process where we discussed general understandings and experiences of participation in mental health (first round) and reflected on the findings from the literature on types of participation (second round); hence we did not use theory-driven interviewing. A thematic analysis of the data generated from the FGDs will enable us to develop broad contextual understanding of this.

- Stage two: why did you propose to extract 1-4 CMOc/paper? This is unusual and typically any included paper you'd extract all CMOc, which will vary depending on the relevance/robustness of the paper to the question.

Response: We have amended the manuscript to not specify a number of CMOs.

- Stage two: a key aspect of a CMOc is that it occurs as a unit. Identifying an enabling mechanisms doesn't say much without the context that triggers it, because this can changes across context, and their relationship produces the outcome.

Response: Thank you for this comment and we agree. We have amended the manuscript to clarify the relationship between the CMO unit.

- The line “how participatory mental health interventions create changes in local context (C), to trigger socio-psychological mechanisms (M) that lead to new behaviour outcomes (O) in communities of interest.”

Response: We think the above response addresses this comment.

Other:

- Please be consistent across the terminology: CMOcs or CMO relationships, IPT, PT or MRT etc.

Response: We have checked the manuscript and ensured consistency of terms.

- In abstract, Intro – ‘respond’ to ‘responds’

Response: Completed.

- In abstract, Methods ‘LIMCs’ to “LMICs”

Response: Completed.

- In abstract, first instance of FGDs please spell out.

Response: Completed.

- In abstract, title LMICs as ‘low-and middle-income’ in introduction, they are ‘low-to-middle income’. For continuity can one term be used throughout.

Response: We have switched to “low and middle” throughout.

- Intro, line 20 – should ‘movement’ be capitalised to be “Movement for GMH”?

Response: Completed.

- Intro, line 33 – comma after “For example”

Response: Completed.

- Intro, Line 37 – awkward ending “...more cost-effective more and ...”

Response: We have removed the second “more”.

- Methods, pg. 9 line 56 “Realist review” to “Realist reviews...”

Response: Completed.

- Please align reporting to, or reference RAMESES 1.

Response: We have stated in the methods section that we align all reporting to RAMESES and referenced accordingly.

Reviewer 2:

Dr. Sonja Klingberg, University of the Witwatersrand

Comments to the Author:

Thank you for the opportunity to review this manuscript. I enjoyed reading this well-developed protocol, and I think both the protocol and, eventually, the review itself will make very valuable contributions to the literature (and hopefully practice!) around participatory approaches to mental health promotion. I found this protocol to be generally well written and very well contextualised,

including an effective overview of theories of participation and a valuable mind map tracing the development of the authors' thinking around the topic.

Response: We thank the reviewer for their kind comments and helpful suggestions below.

Specific comments and questions:

1. Apart from the limited timeframe, it is not fully clear to me why this extensive process, including valuable stakeholder engagement, "only" amounts to a rapid realist review and not a full realist review but the process is otherwise clearly described. No action needs to be taken in response to this comment but it does seem to me like you are doing more or less the same amount of work as a full realist review would require.

Review: Thank you for this comment. We realise this is more appropriately titled a "realist review" and have amended the manuscript accordingly.

2. My main feedback concerns the initial programme theory as I am finding the text version of it a little difficult to follow. It seems to suggest participatory approaches are a condition of effective interventions? Does this imply something specific about meaningful outcomes as a result of meaningful participation, i.e. would positive outcomes of 'not meaningfully' participatory interventions not be as meaningful (e.g. not transformative or empowering), or are you theorising that only meaningful participation will actually result in effective interventions? This is naturally something the review itself will clarify and refine but it is difficult to follow the argument here. It would be helpful to clarify the text description of the theory as Figure 3 does not express it in quite such specific terms. It may also be relevant to consider Abimbola's interesting paper "Beyond positive a priori bias: reframing community engagement in LMICs." in this context. However, to be clear, I am not saying that the initial programme theory itself needs to change – it may just need to be rephrased and expressed more explicitly.

Response: Thank you for this comment. We have tried to do this through including some more detail of the 'because' aspect, in line with reviewer 1 comments, and also by highlighting that things could work in the absence of meaningful participation, and saying what that work/impact could result in.

3. The relationship between Context and the other constructs in Figure 3 is not fully clear – is it a cross-cutting dimension feeding into all the others equally?

Response: We have revised figure 3 (see response to the next point).

4. It also seems to me based on Figure 3 that you are proposing that Interventions are included in the CMO configurations as a separate construct as (CMIOc) as opposed to being expressed through mechanisms. It may well be very relevant to do so, but this is not described in the steps of the review so this aspect could also be clarified or elaborated even if just to say that it is a possibility and the review itself will determine its relevance.

Response: Thank you for these comments, we have revised the figure – moving these aspects out of line from others and hope that this makes it clearer. We have also edited the methods section of the manuscript to clarify this.

5. Very minor comment: "Qualitative" is listed twice under inclusion criteria.

Response: We have amended accordingly

VERSION 2 – REVIEW

REVIEWER	Klingberg, Sonja University of the Witwatersrand, SAMRC/Wits Developmental Pathways for Health Research Unit (DPHRU)
REVIEW RETURNED	22-Feb-2022
GENERAL COMMENTS	Thank you, I am happy with how my comments and questions have been addressed. The initial theorising is now much easier to follow and I look forward to seeing the full review.